# PLAUSIBLY DENIABLE ENCRYPTION
# WITH LARGE LANGUAGE MODELS

## ABSTRACT

We present a novel approach for achieving plausible deniability in cryptography by harnessing the power of large language models (LLMs) in conjunction with conventional encryption algorithms. Leveraging the inherent statistical properties of LLMs, we design an encryption scheme that allows the same ciphertext to be decrypted with any key, while still yielding a plausible message. Unlike established methods, our approach neither relies on a fixed set of decoy keys or messages nor introduces redundancy. Our method is founded on the observation that language models can be used as encoders to compress a low-entropy signal (such as natural language) into a stream indistinguishable from noise, and similarly, that sampling from the model is equivalent to decoding a stream of noise. When such a stream is encrypted and subsequently decrypted with an incorrect key, it will lead to a sampling behavior and will thus generate a plausible message. Through a series of experiments, we substantiate the resilience of our approach against various statistical detection techniques. Finally, although we mainly focus on language models, we establish the applicability of our approach to a broader set of generative models and domains, including images and audio.

## 1 INTRODUCTION

In the realm of cryptography, the notion of plausible deniability refers to encryption techniques where an adversary cannot definitively establish the mere existence of an encrypted file or communication, as they are unable to prove the presence of the original, unencrypted data (Canetti et al., 1997; Klonowski et al., 2008). This means that, even when forced to reveal the decrypted message, one can still convincingly claim ignorance about its contents, as illustrated in Figure 1.

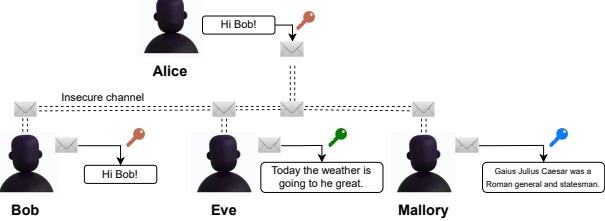

Figure 1: Imagine a scenario where Alice is determined to share a confidential message with Bob, through a possibly insecure channel. She then sends the encrypted file to Bob, who extracts the hidden message using the secret key. Even if someone intercepts the transmitted encrypted message and coerces Bob into providing the key that leads to its decryption, Bob can provide a (randomly chosen) key as the supposed correct one. The new key leads to a distinct yet entirely believable decryption of the message – one that appears as plausible and realistic.

The majority of real-world applications of plausible deniability in cryptography focus on situations where the existence of encrypted material is entirely refuted, such as rejecting the existence of an encrypted partition on a hard drive – a widely recognized function in encryption suites like True-Crypt (Broz & Matyas, 2014) – or *steganography*, where the encrypted content is concealed within another message (e.g. an image), potentially introducing redundancy. A more intriguing challenge lies in designing an algorithm that allows a single ciphertext – or encrypted communication – to

be decrypted into multiple possible plaintexts depending on the selected key. Several schemes have been proposed to address this problem (Trachtenberg; Koç, 2009; Ferguson et al., 2011; LibreCrypt), but they often introduce redundancy and rely on a fixed number of decoy keys, each associated with a decoy message. An important question arises: *Given a white-box knowledge of the algorithm, what if the attacker coerces the defendant into revealing all possible keys?*

In this work, instead, we focus on a more general formulation of the problem, where the *ciphertext* can be decoded to a valid plausible message using *any* key. We do so by leveraging the powerful statistical properties of large language models coupled with conventional encryption algorithms such as Advanced Encryption Standard (AES) (Daemen & Rijmen, 2002; Nechvatal et al., 2001).

Our idea is inspired by the following principles, which we explore in-depth in subsequent sections:

- A language model, being an explicit probabilistic model, allows for two operations:
  - Encoding (or *compression*): based on the observation that different symbols (tokens) appear with varying probabilities, this operation compresses a low-entropy message (such as natural language) into a compact signal that appears indistinguishable from white noise.
  - Decoding (or *sampling*): this operation converts a stream of white noise into a low-entropy message, which represents the typical strategy for generating text in LLMs.
- In encryption algorithms, when a *ciphertext* is decrypted using the wrong key, the resulting *plaintext* is white noise[1].
- When the *plaintext* is encoded using an LLM and subsequently encrypted, it can be decrypted with any key and still produce a plausible message when decoded with the LLM (sampling behavior).
- Regardless of the key used for encryption/decryption, when a strong model is used, the *encoded plaintext* is indistinguishable from white noise, rendering statistical detection tests ineffective.

## 2 BACKGROUND AND RELATED WORK

**Compression**    Compression and model-driven prediction demonstrate a fundamental correspondence, sharing the common objective of reducing information redundancy. Compressors aim to efficiently encode data by capturing key patterns (Rahman & Hamada, 2021; Valmeekam et al., 2023), while predictors seek to anticipate future observations by leveraging past information (Kolmogoroff, 1933). The source coding theorem (Shannon, 1948) states that an optimal encoder compresses a message to an expected length in bits equal to the $\log_2$-likelihood of the statistical model (Delétang et al., 2023), essentially reducing the message to a shorter stream that appears sampled from a uniform distribution (white noise). Recently, LLMs have proven to be highly successful statistical models (Brown et al., 2020; Touvron et al., 2023) that effectively capture the intricate nuances of the underlying distribution. These models, tokenize the text into bytes/subwords (Yu et al., 2023; Edman et al., 2022; Sennrich et al., 2015), and are trained towards maximizing the conditional probability of the next token, given the preceding context. This makes them an ideal choice for compressing text and its associated distribution.

**Entropy coding**    At the heart of lossless compression lies the task of transforming a sequence of symbols into a succinct bit sequence, all while preserving the ability to fully reconstruct the original symbol sequence. A variety of different techniques achieve this, including Huffman coding (Huffman, 1952), arithmetic coding (Pasco, 1976; Rissanen, 1976), and asymmetric numeral systems (Duda, 2009), to name a few. Huffman coding capitalizes on the uneven probabilities governing symbol occurrences, assigning bitstreams of varying lengths to symbols based on their frequency in the data. Shannon's source coding theorem establishes the limit $L$ on possible data compression as $L \geq \mathcal{H}(\rho)$, where $\mathcal{H}(\rho) := \mathbb{E}_{x \sim \rho}[-\log_2 \rho(x)]$ and $\rho$ is the distribution of the tokens.

**ML and Cryptography**    As a fundamental pillar of information security, cryptography (Stamp, 2011; Feistel, 1973) plays a crucial role in safeguarding sensitive data and communications. Recent advancements have been significantly influenced by machine learning. These innovations span a wide spectrum of applications, ranging from general cryptographic techniques to more specialized areas such as differential cryptanalysis and distinguishing attacks. One major area of interest is *differential cryptanalysis*, studying how input perturbations are propagated to the produced output. Emerging research explores the synergy between machine learning and cryptography, e.g. in

---

[1]This is also the basis for many cryptographically-secure random number generators.

devising attacks to distinguish cryptographic text, leveraging potential distribution shifts in various proposed encryption algorithms (Gohr, 2019; Wenger et al., 2022). This highlights the evolving landscape of security paradigms, reflecting an exciting avenue for the future of data protection.

**Plausible deniability**    Cryptography attempts to make information unintelligible (Walton, 1996). Techniques for concealing information (Petitcolas et al., 1999) include steganography (Channalli & Jadhav, 2009), watermarking, and the practice of embedding messages within other network traffic (Rivest et al., 1998). Plausible deniability does not preclude the interception of a message by an adversary; rather, it provides the means to disavow its true meaning, adding another layer of defence against privacy breaches involving sensitive data. It offers resilience against coercion attempts, wherein adversaries demand access to the plaintext data concealed within ciphertext (see Canetti et al. (1997); Dürmuth & Freeman (2011) for a formal definition). Conventional encryption schemes do not inherently provide deniability. Although a few approaches proposed how to introduce deniability, they are limited in their capabilities, e.g., LibreCrypt, or require substantial computational resources and lead to considerably lengthier messages to be transmitted (Stevens & Su, 2023).

**Sampling**    In the context of autoregressive language modeling, the selection of an appropriate sampling algorithm plays a crucial role in generating text that is both coherent and diverse (Holtzman et al., 2019; Meister et al., 2023; Hewitt et al., 2022). Among other proposed techniques, are temperature scaling, top-$k$ sampling, and top-$p$ sampling (also known as Nucleus sampling (Holtzman et al., 2019)). Current trends also underscore the use of techniques such as prompting (Wei et al., 2022) and fine-tuning (Ouyang et al., 2022) to influence the generative process of LLMs, all of which attempt to manipulate the probability distribution predicted by the model.

## 3 METHOD

We provide an overview of our methodology in Figure 2. Initially, the involved parties in the communication establish a consensus on a model and exchange a shared encryption key (*shared-key* scheme). Our detailed security scheme is outlined in Appendix A.1, and its correctness is assessed in Appendix A.2. The core of our approach begins with the tokenization of a given text string, transforming it into a sequence of discrete tokens. The selected model operates in an autoregressive manner, predicting the probability distribution of every token based on the preceding ones in the sequence. From the extracted cumulative distribution, the input text is transformed into a series of codes. In the decoding phase, we apply a reverse procedure, where, instead of sampling during generation, we use the transmitted codes to guide the selection of the tokens.

### 3.1 BASE METHOD

Before explaining our final method, we introduce a simplified – yet more interpretable – variant that illustrates some important concepts related to sampling from LLMs.

**Quantization**    Since common encryption algorithms work with discrete numbers, we work with quantized probability distributions. This also guarantees that all operations are deterministic and are not affected by rounding or accumulation errors of floating-point numbers while simplifying the implementation of the algorithm. More specifically, we quantize each probability distribution (model output) to $N = 2^k$ bins, such that the cumulative probability sums to $N$. Possible choices of $k$ are those that are concurrently byte-aligned and divisors of the block size of the adopted encryption algorithm (128 bits for AES), e.g. $k = 8, 16, 32$. The choice of $k$ has important implications, which we discuss in one of the next paragraphs.

**Sampling/decoding**    For simplicity, we first explain how *decoding* is performed, and establish its connection with *sampling* in LLMs. During generation, for each token fed through the language model, the model explicitly predicts the probability distribution of the next token conditioned on all previous tokens. Generating text involves sampling from this probability distribution and repeating the process in a feedback fashion. From a more technical perspective, sampling is implemented using the *inverse transform sampling* algorithm, which involves *(i)* computing the c.d.f. of the probability distribution, *(ii)* sampling a random number $t$ from $\mathcal{U}(0, 1)$ (or, in our discrete case, $\mathcal{U}\{0, 2^k - 1\}$ inclusive), and *(iii)* finding the bin in which $t$ falls through binary search. We call this process *decoding* when it is carried out deterministically using a stream of numbers $t_i$ (which appear as random, but may or may not be the result of a random process) provided externally.

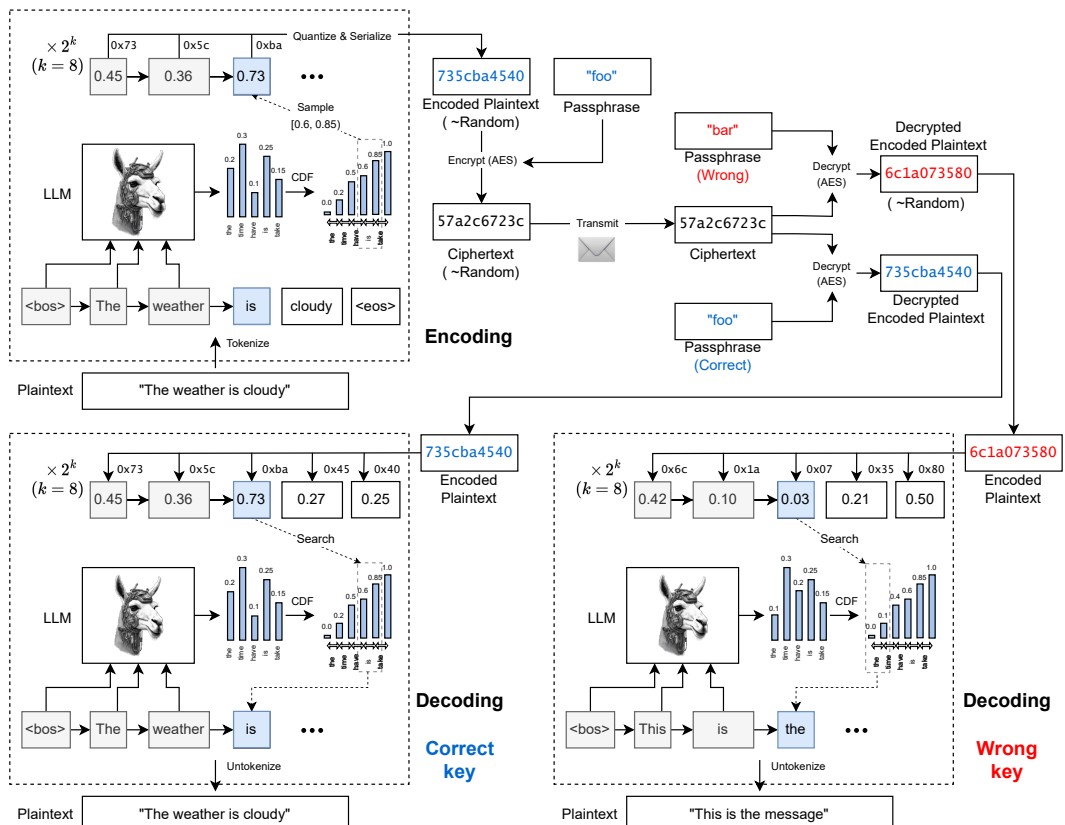

Figure 2: The pipeline for the base method. Using the probability distributions predicted by the model, the plaintext is encoded into a stream indistinguishable from noise, and subsequently encrypted. In this fictitious example, we use $k = 8$, i.e. each token is encoded using one byte. When the ciphertext is decrypted with the correct key and decoded, the original message is recovered. Conversely, decrypting the ciphertext with the wrong key results in a random stream that will still lead to a plausible message when decoded. For simplicity, implementation details such as headers, padding, and initialization vectors (IV) are omitted. The depicted pipeline is also applicable to the compressed encoder, in which the sampling/search mechanism is replaced with an entropy coder.

**Encoding** The *encoding* process creates a stream of numbers $t_i$ that decode into the original message, and which, under optimality assumptions, is indistinguishable from white noise. For each token $i$ in the plaintext and its corresponding probability distribution obtained by querying the model, we *(i)* compute its c.d.f., *(ii)* find the lower bound $t_i^{[l]}$, as well as its upper bound $t_i^{[u]}$ (i.e. the lower bound of the next token in the vocabulary), and *(iii)* sample $t_i$ from $\mathcal{U}\{t_i^{[l]}, t_i^{[u]} - 1\}$ (inclusive).

**Encryption/decryption** The *encoded plaintext* (stream of $t_i$) is then trivially encrypted using a standard algorithm such as AES (with some caveats described in subsection 3.5). When decrypted with the correct key, the message is decoded into the original plaintext without loss. When decrypted with the wrong key, the resulting stream of $t_i$ will be random, exhibiting a sampling behavior when decoded using the language model (and thus generating a random but plausible message).

**Limitations and choice of** $k$ The scheme presented is statistically simple to analyse but suboptimal from an information-theoretic standpoint. The algorithm's behavior is influenced by the quantization granularity $k$, where larger values of $k$ provide an accurate probability distribution representation but increase the encoded message redundancy, leading to longer messages. Smaller values of $k$ result in more compact messages but may introduce quantization errors or cause unlikely tokens to vanish entirely from the probability mass, rendering their encoding *unsatisfiable* (this case can however be detected and an error returned). In our experiments, a sensible choice is $k = 32$, encoding each token as a 4-byte integer.

## 3.2 COMPRESSED ENCODING

In this section, we propose a more principled encoding/decoding scheme that is based on *compression* and is optimal in the sense that it *does not introduce redundancy nor does it depend on the choice of a quantization granularity $k$*. To this end, we make use of variable-length coding, and more specifically Huffman trees (Huffman, 1952), which represents the established way of performing compression in standard compression algorithms, e.g. DEFLATE (Oswal et al., 2016). Guiding this scheme using the probability distributions obtained from the language model allows us to design an encoder that can be used as a drop-in replacement to the base algorithm. Furthermore, an interesting by-product of this formulation is that the message is compressed in a lossless fashion in addition to being encrypted[2].

**Language models as compressors** Consider a vocabulary of 4 tokens $A$, $B$, $C$, $D$, occurring with probabilities $0.5, 0.25, 0.125, 0.125$ respectively (which may as well be the output of a fictitious model). Each of these symbols can naively be encoded using 2 bits per token, e.g. 00, 01, 10, 11 respectively. However, an optimal variable-length codebook (i.e. a Huffman code) would encode these as 0, 10, 110, 111, achieving an optimal compression rate of 1.75 bits per token (i.e. the entropy of the original probability distribution). By obtaining these probabilities using a language model sequentially, it is possible to design an encoder that compresses natural language into a stream of bits which, under optimality assumptions, is (again) indistinguishable from noise. Modern language models are trained with the goal of predicting the next token distribution, and as a result, they are explicitly designed to compress the text they have been trained upon. It has also been shown that this task scales well to larger models, which are able to achieve increasingly higher compression rates as measured by perplexity metrics (Kaplan et al., 2020; Hoffmann et al., 2022; Touvron et al., 2023).

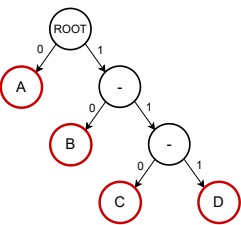

Figure 3: The Huffman tree corresponding to the example. A Huffman tree can be generated in $\mathcal{O}(n \log n)$, where $n$ represents the number of unique symbols, i.e. tokens, in the text vocabulary.

**Encoding** Under this scheme, for each token position $i$, we construct a *Huffman tree* using the probability distribution predicted by the previous token, and we append the code corresponding to the token $i$ to the stream. If padding is needed (e.g. to align the stream to a byte or a predefined block size), we append random bits (see also implementation details, subsection 3.5, for the reasoning).

**Decoding** Decoding is trivially achieved by running the process in reverse. If a wrong key is used to decrypt the ciphertext, the resulting bit stream will be random, but it is guaranteed that it can be decoded successfully, as Huffman trees are *full binary trees* (each node is either a leaf or has 2 children). As in the baseline algorithm, this will result in a sampling behavior (i.e. random text generation). However, in case the wrong key is used, the stream might terminate early (leaving the last symbol partially decoded), which provides a trivial detection mechanism. Therefore, random padding is crucial to elude such detection attempts. Due to its importance (including in the base formulation), we discuss this aspect in the implementation details, subsection 3.5.

**Limitations** Although Huffman codes are optimal among schemes that encode each symbol separately, they achieve the optimal entropy only if the input probability distribution is *dyadic*, i.e. it can be expressed as $p(i) = 2^{-x_i}$ (where $x_i$ is a positive integer). If the distribution diverges too much from these "dyadic points", detection will be facilitated as the resulting encoded stream will not be indistinguishable from noise. However, this is typically a concern only for small vocabularies, which is not the case for language models (which often comprise tens of thousands of tokens). Although there exist more optimal coding schemes, such as *arithmetic coding*, these are very slow, complex to implement, and are therefore beyond the scope of our work.

---

[2]Modern LLMs achieve compression rates in the order of 4 bits per token (depending on the tokenizer). This can also be computed as $\log_2(\text{PPL})$, where PPL denotes the normalized perplexity of the message. It is therefore clear that the choice of $k = 32$ bits per token proposed earlier is highly redundant.

### 3.3 Trivial extensions

**Sampling schemes** Our approach can be trivially extended to various sampling strategies designed for language models, such as top-$k$ sampling, top-$p$ sampling (a.k.a. Nucleus sampling (Holtzman et al., 2019)), and temperature scaling. The only caveat is that these settings must be transmitted as part of the language model or the ciphertext, or simply agreed to before transmitting the message. It is also worth noting that these sampling strategies, if set to excessive strength, may render statistical detection easier or cause some tokens to be *unsatisfiable* if they are excluded from the probability mass (this however only affects the base algorithm), as we show in section 5.

**Prompting** If a wrong key is used, random sampling from an unconditional distribution might return messages that, although grammatically valid, are too unrelated to the context expected by the adversary. For instance, if the adversary expects the location or time of a meeting, most likely a randomly generated news article will not represent a good fit for a decoy message. Fortunately, our algorithm can be trivially extended with a prompting mechanism that provides a context to the model, such as "`Location of the meeting. City: New York. Address: `", which needs to be transmitted in unencrypted form as part of the ciphertext. This context-setting mechanism plays a pivotal role in guiding the model's output towards producing responses that align with possible expected responses. In our example, the prompt instructs the model to generate content that pertains to the location and address of the meeting in New York, making it significantly less likely to generate irrelevant or generic text (Wei et al., 2022).

**Chat variant** Similarly, employing a chat version of a language model, it is possible to encrypt entire conversations between two or more parties and ensure that these decrypt into plausible conversations if the wrong key is used. Prompting can again be used to guide the context.

### 3.4 Connection to other domains

The proposed idea can be applied to any explicit generative model, including autoregressive models for audio – e.g. WaveNet (Oord et al., 2016)) – and images – e.g. PixelCNN (Van den Oord et al., 2016), PixelRNN (Van Den Oord et al., 2016), ImageGPT (Chen et al., 2020) – as well as variational autoencoders (VAEs) (Kingma & Welling, 2013). The idea can also be generalized (with some variations) to other classes of generative models, including GANs (Goodfellow et al., 2014) and diffusion models (Ho et al., 2020). As an example for the latter, the uniform noise produced by the decryption algorithm can be converted to Gaussian noise through a Box-Muller transform (Box & Muller, 1958), and then used as a seed for the denoising process in conjunction with a deterministic sampler such as DDIM (Song et al., 2020). As part of our work, to provide a further validation of the algorithm, we apply our approach to ImageGPT and show some qualitative examples on images in section 5.

### 3.5 Implementation details

**Encryption and padding** For encryption, we use AES-256 in CBC mode. The algorithm is primed with a random *initialization vector* (IV) which is included in the *ciphertext* in unencrypted form. In block cipher algorithms, it is common to pad the *plaintext* message to a predefined block size (128 bits for AES), both for technical reasons and to avoid leaking information about the length of the message. In our case, the latter aspect is even more important as the original message may include an explicit sentence terminator (`<eos>` token), which also needs to appear when the message is decrypted with the wrong key (otherwise, detection would be trivial). Therefore, the message needs to be padded sufficiently to ensure that the sentence terminator is sampled even if a wrong key is used. We discuss potential padding strategies in the Appendix A.3. In all formulations, we pad the message with random bits, which results in a sampling behavior and eludes detection mechanisms. For our experiments on ImageGPT, padding is not needed as we always generate images of fixed size.

**Deterministic model evaluation** To ensure that the algorithm is correct, the entire pipeline (especially the model evaluation) needs to be deterministic. This is not the case for floating-point tensor operators in GPUs, which are subject to non-deterministic accumulation. For this reason, we recommend using quantized models (either as integers or fixed-point precision), or running the model through a device that is guaranteed to exhibit per-sample determinism regardless of batch size.

| | | | IMDb | | | | | | Twitter | | | | | |
| | | | Correct key | | | Wrong key | | | Correct key | | | Wrong key | | |
| Encoder | Sampling | Model | Freq | Corr | Ppl | Freq | Corr | Ppl | Freq | Corr | Ppl | Freq | Corr | Ppl |
|---|---|---|---|---|---|---|---|---|---|---|---|---|---|---|
| Base | \multicolumn Dummy distribution | | **1.000** | **0.983** | **1.000** | 0.011 | 0.009 | 0.011 | **0.934** | **0.535** | **1.000** | 0.007 | 0.007 | 0.007 |
| | top-$p$ | GPT2-xl | 0.255 | 0.182 | **1.000** | 0.009 | 0.009 | 0.000 | 0.011 | 0.041 | **0.991** | 0.009 | 0.002 | 0.006 |
| | | LLaMa2-7B | 0.020 | 0.083 | **1.000** | 0.009 | 0.011 | 0.006 | 0.023 | 0.023 | **1.000** | 0.017 | 0.013 | 0.009 |
| | top-$k$ | GPT2-xl | 0.045 | 0.009 | **0.891** | 0.000 | 0.009 | 0.018 | 0.006 | 0.011 | **0.789** | 0.013 | 0.004 | 0.021 |
| | | LLaMa2-7B | 0.006 | 0.006 | **0.772** | 0.011 | 0.017 | 0.009 | 0.018 | 0.007 | **0.816** | 0.016 | 0.020 | 0.005 |
| | Unbiased | GPT2-xl | 0.009 | 0.009 | 0.037 | 0.000 | 0.018 | 0.018 | 0.019 | 0.017 | 0.110 | 0.006 | 0.002 | 0.006 |
| | | LLaMa2-7B | 0.011 | 0.017 | 0.028 | 0.003 | 0.023 | 0.017 | 0.009 | 0.011 | 0.232 | 0.020 | 0.005 | 0.007 |
| Compressed | Dummy distribution | | 0.018 | 0.011 | **1.000** | 0.011 | 0.005 | 0.010 | 0.011 | 0.004 | **1.000** | 0.015 | 0.002 | 0.011 |
| | top-$p$ | GPT2-xl | 0.050 | 0.030 | **1.000** | 0.035 | 0.020 | 0.015 | 0.019 | 0.002 | **0.973** | 0.011 | 0.017 | 0.004 |
| | | LLaMa2-7B | 0.101 | 0.028 | **0.987** | 0.018 | 0.006 | 0.009 | 0.007 | 0.005 | **0.984** | 0.011 | 0.005 | 0.000 |
| | top-$k$ | GPT2-xl | 0.052 | 0.022 | **0.599** | 0.004 | 0.004 | 0.000 | 0.013 | 0.006 | **0.666** | 0.017 | 0.011 | 0.008 |
| | | LLaMa2-7B | 0.160 | 0.044 | 0.114 | 0.006 | 0.006 | 0.010 | 0.013 | 0.009 | **0.637** | 0.022 | 0.015 | 0.020 |
| | Unbiased | GPT2-xl | 0.053 | 0.019 | 0.148 | 0.004 | 0.011 | 0.000 | 0.019 | 0.008 | 0.063 | 0.017 | 0.006 | 0.006 |
| | | LLaMa2-7B | 0.183 | 0.050 | 0.403 | 0.014 | 0.009 | 0.024 | 0.013 | 0.009 | 0.113 | 0.018 | 0.004 | 0.005 |

Table 1: Main experiments on the IMDb dataset and tweets, under multiple variations of sampling mechanisms and models. We report the fraction of samples whose corresponding null hypothesis is rejected ($p < 0.01$, meaning that each experiment should exhibit an expected false-positive rate of $0.01$), under the three proposed statistical tests (frequency, correlation, and perplexity). A value close to the expected false-positive rate of $0.01$ signifies that the statistical test fails to detect that the true key has been used. As a sanity check, we run the same experiments on messages decoded using the wrong key, and observe results in line with expectations. We highlight in **bold** experiments where more than half of samples are detected as being encrypted with the given key. To satisfy the assumptions of the tests, we select only samples consisting of at least 20 tokens.

## 4 DETECTION

Under both formulations, the following properties need to be satisfied to elude detection: *(1)* the *encoded* plaintext (i.e. after encoding but before encryption) must be indistinguishable from white noise (i.e. a stream sampled from a uniform distribution); *(2)* the plaintext message must appear as if it was sampled from the model. These properties are satisfied if the language model is representative of natural language, or likewise, if the message falls within the domain on which the language model was trained. Otherwise, the resulting bias can in principle be detected using statistical tests. While these are not meant to prove the adherence of our method to the security model proposed in Appendix A.1, they represent a useful tool for assessing the goodness of fit of the model, as well as a sanity check. We refer the reader to Appendix A.2 for a more detailed discussion of correctness.

The first property can be empirically tested using standard techniques to assess the quality of random number generators (Marsaglia, 2008). We employ two tests, a *frequency test* to evaluate whether all symbols occur with the same probability, and a *correlation test* to verify whether symbols are sequentially uncorrelated. For the frequency test, we perform a $\mathcal{X}^2$ (chi-squared) test on a sliding window of 8 bits over the encoded plaintext, testing the null hypothesis that the observed frequencies have been sampled from a uniform distribution. For the correlation test, we employ a *runs test* (Bradley, 1960) over individual bits, which tests the null hypothesis that samples are uncorrelated. These tests are efficient as they do not require querying the model, but detecting a bias with a sufficient significance level might only be feasible with a particularly weak model.

Evaluating the second property is less trivial, as the task is equivalent to detecting whether a given text was (or was not) generated using a language model. There have been a plethora of works in this area (Guo et al., 2023; Clark et al., 2021; Mitchell et al., 2023; Kirchenbauer et al., 2023; Sadasivan et al., 2023; Tang et al., 2023). In this work, we adopt a statistical approach. Each message has an associated *information content* expressed in bits, i.e. the theoretical length of the message if it was compressed optimally. For a large number of tokens (e.g. $N > 20$), the distribution of this measure converges to a Gaussian distribution with a mean and variance determinable analytically from the model outputs. After determining these parameters, we apply a two-tailed test to infer whether the message falls outside the mass of the distribution, meaning that it is unlikely to have been generated by the model. We informally name this test *perplexity test* since the term is more familiar in the literature, although it is not directly based on the common perplexity measure used to evaluate LLMs. We provide more details and assumptions in the Appendix A.4.

## 5 EXPERIMENTS

**Experimental setting** To validate our proposed statistical tests on real-world data, we conduct a series of experiments using two publicly available LLMs: GPT2-xl (1.5B parameters) (Radford

et al., 2019) and LLaMA2-7B (7B parameters) (Touvron et al., 2023). We also evaluate the impact of various sampling techniques used in the literature: in addition to unbiased sampling, we assess top-$k$ sampling ($k = 50$) as well as a popular setting whether both temperature scaling and top-$p$ sampling are used ($\tau = 0.8$, $p = 0.9$). To avoid setting probabilities to exactly zero (thus making some tokens *unsatisfiable*), we smooth the biased distribution by mixing it with the original distribution ($\mathbf{p} = (1-t)\,\mathbf{p}_{\text{biased}} + t\,\mathbf{p}_{\text{original}}$, $t = 0.4$). Furthermore, to assess the reactivity of the tests, we include a very simple baseline consisting of a randomly generated distribution (*dummy distribution*). Each experiment is carried out using both our proposed encoders: the base encoder, which encodes each token as a fixed-length symbol, and the compressed encoder, which encodes tokens using variable-length symbols. We conducted our experiments using two datasets that contain colloquial text: IMDb reviews and Twitter messages (see A.5). This selection was made to mirror the authentic distribution of textual content and different communication styles. Finally, we also present some qualitative results on images encoded using the publicly available version of ImageGPT-large (Chen et al., 2020), which operates at $32 \times 32$ resolution with pixel-level tokenization.

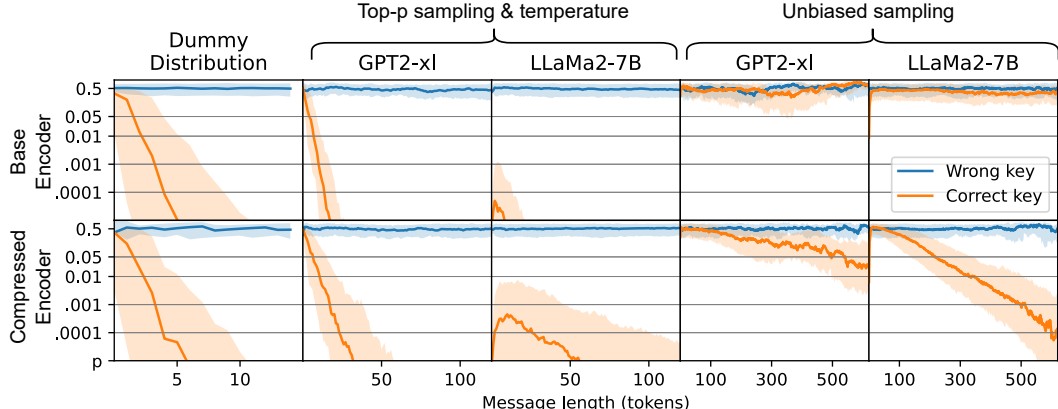

Figure 4: Distribution of the $p$-values of the perplexity test, as a function of the message length (IMDb dataset). Each line depicts the median $p$-value across different texts in the dataset, whereas the shaded area describes the 25th and 75th percentiles. As a sanity check, we also show the $p$-values corresponding to messages decoded using the wrong key, which are expected to float around $p = 0.5$ regardless of length.

**Results** We show our main quantitative results in Table 1, where we run all three statistical tests for each model and sampling setting. Under the considered datasets, we observe that the model-free tests (frequency test and runs test) are effective only on severely mismatched distributions such as the dummy one, but fail to detect when the correct key is used in LLM experiments. The perplexity test, which uses the model outputs, appears to be more sensitive in our experiments: it successfully detects that the correct key has been used in the biased settings (top-$p$ and top-$k$ sampling), but it is still unreliable in the unbiased setting. This suggests that modern LLMs learn an accurate statistical representation of natural language, whose bias is intricate to pick up even by the most sensitive statistical tests. We also observe that the portion of successful detections increases when the compressed encoder is used, as the tests pick up on the slight bias introduced by the Huffman encoding. This is however expected to play a significant role only on lengthy messages, which brings us to our next experiment: analysing the dependency of the perplexity test on the length of the message. We show our findings in Figure 4, where we plot the distribution of $p$-values as a function of message length. We observe that using a dummy distribution leads to a successful detection in as few as 5–10 tokens, and that biased sampling techniques can also be detected fairly quickly (20–50 tokens). This is due to the fact that top-$p$ and top-$k$ sampling set some probabilities to zero (or close to zero in our smoothed variant), providing an easy cue for the test when these tokens appear in a message. In the unbiased sampling scenario, we observe that the base encoder appears to be undetectable even for long sequences (500+ tokens), whereas the statistical bias of the compressed encoder is detected with a sufficient significance level after 300–400 tokens. Surprisingly, the bias is detected faster on LLaMA2, despite this model being larger and more recent than GPT2. We attribute this effect to the smaller size of the vocabulary of LLaMA (32k tokens *vs* 50k tokens of GPT-2), which makes the Huffman encoding less effective.

**Qualitative examples** We present qualitative results when decrypting text using wrong keys in Table 2 and Table 3 in the Appendix. Finally, we show a set of qualitative examples on ImageGPT in Figure 5, where we can also directly visualize the encoded and encrypted representations, further validating our assumptions.

| Prompt | Correct Message | Decoding 1 | Decoding 2 | Decoding 3 |
|---|---|---|---|---|
| Location of the meeting. City: New York. Address: | 48 Street and, Park Ave | 44 Wall Street, New York, NY 10005 | 15 W 41st St, New York, NY 1 | 100 Broadway, New York, NY 10005 |
| Hey, did you hear we are all gathering tonight for: | April's suprise party! | The Karaoke Contest! | A LIVE WEBINAR | "Crazy Cat Lady Appreciation Night" |

Table 2: Qualitative examples of decoding messages that were decrypted with the wrong keys. Statistical properties of the language model that we are using – LLaMa2-7B in this case – ensure that decoded messages result in coherent and plausible texts.

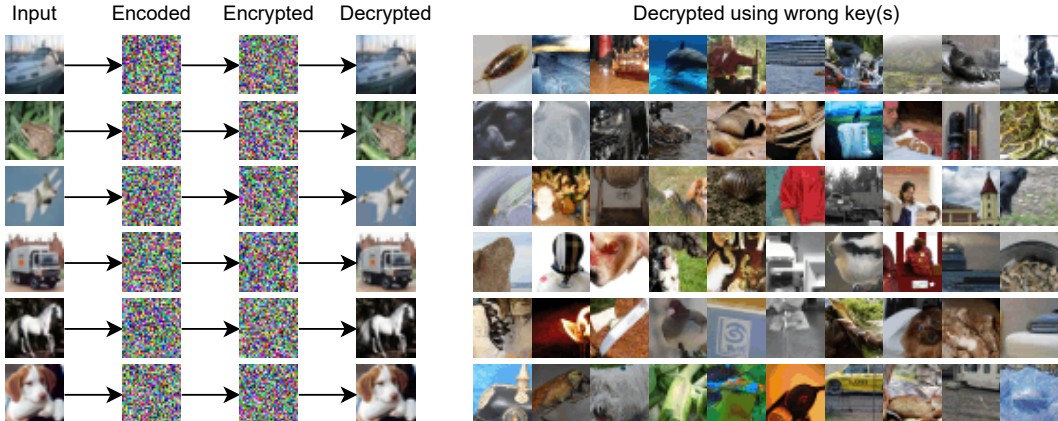

Figure 5: Example of applying our method to ImageGPT on RGB images of size $32 \times 32$ from CIFAR-10. We use the base encoder with $k = 24$ bits/pixel, which allows for easy visualization of the *encoded plaintext* (second column) and the *ciphertext* (third column) by mapping each encoded symbol to a 24-bit RGB value. As expected, the encoded image (prior to encryption) is perceptually indistinguishable from noise, providing a further visual validation of the algorithm. While the original image is reconstructed correctly when decrypted with the proper key, decrypting it with a wrong key leads to a random plausible image.

## 6 CONCLUSION

We proposed a framework for achieving plausibly deniable encryption using LLMs. Our approach combines the statistical properties of these models with a layer of standard encryption, providing a way to generate plausible messages when the ciphertext is decrypted using any wrong key. We proposed a minimally redundant scheme based on Huffman coding which achieves compression in addition to encryption, and devised a framework to assess the robustness of these representations against statistical tests. Finally, we demonstrated a set of qualitative examples on both text and images. In future work, we would like to extend our approach to other domains such as audio, and design a broader suite of statistical tests to find potential attack vectors.

**Limitations** As mentioned, the model weights constitute part of the algorithm and need to be agreed upon and stored by all parties. Furthermore, since the language model must be evaluated sequentially, the decoding process can be slow. Encoding on the other hand can still be parallelized. This limitation is however expected to play a lesser role in the future, thanks to more efficient implementations as well as quantized variants of these models that can also be run on embedded devices. This is also corroborated by recent advances in specialized hardware for deep learning models (Shahid & Mushtaq, 2020; Dhilleswararao et al., 2022).

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

# A APPENDIX

## A.1 SECURITY MODEL

We describe our set of assumptions and our adopted security model. Our scheme is closest to the *shared-key* deniable encryption scheme proposed in *Definition 4* of Canetti et al. (1997), in which the sender and receiver agree upon a shared secret key $s$ prior to exchanging information, but presents some important differences w.r.t. the deniability formulation. We also assume a *white-box* threat model, in which a potential attacker has access to the algorithm and the encoder weights. Denoting our communication protocol as $\pi$, our security model needs to satisfy the following properties:

**Correctness:** when the correct key is used, the receiver must decrypt the original message without loss. Although the definition in Canetti et al. (1997) allows for a negligible error tolerance, here we assume a lossless scenario (after accounting for tokenization or quantization).

**Security:** the protocol must be secure against eavesdropping. More formally, denoting the communication between parties as $\text{COM}_\pi$ (i.e. the transmitted *ciphertext*), for any two random messages $m_1$, $m_2$ and random key $s$, it must follow that $\text{COM}_\pi(m_1, s)$ and $\text{COM}_\pi(m_2, s)$ are *indistinguishable* in the distributional sense.

**Distributional Deniability:** when an alternative key is used, the decryption on the receiver's end must result in a plausible message. Specifically, for any message $m_1$ (the original message) and keys $s_1$, $s_2$ chosen at random, let $c = \text{COM}_\pi(m_1, s_1)$ be the communication of $m_1$ encrypted with the correct key $s_1$, and let $m_2$ be the decryption of $c$ with the alternative key $s_2$. It must follow that the distributions of $m_1$ and $m_2$ are *indistinguishable*.

Throughout this work, we rely on the definition of computational indistinguishability provided in *Definition 1* of Canetti et al. (1997), according to which two ensembles of distributions should be indistinguishable by a polynomial-time adversary.

As opposed to the deniability property assumed by Canetti et al. (1997), in which $m_2$ can be chosen arbitrarily (and the corresponding decoy key $s_2$ is derived through a "faking" function), our formulation allows for less control over $m_2$ and simply assumes that it is *plausible* (i.e. it is drawn from the distribution of real messages). This somewhat weaker assumption is however counterbalanced by a greater flexibility regarding the scenarios in which our method can be applied. For example, our scheme can be implemented in a way that does not introduce redundancy, nor does it require a key as long as the message (a well-known limitation of one-time pads). Furthermore, our formulation allows the receiver to choose *any* decoy key $s_2$, and as late as at the time of attack. Finally, control over $m_2$ can also be improved by prompting, which we study in subsection 3.3.

Another important consideration is that, although our scheme applies to any distribution of messages $m$ (e.g. a known multinomial distribution), in practice the true distribution of $m$ may be unknown, such as in the case of natural language or images. Therefore, we assume the availability of a model that is representative of the true distribution of messages. If this were not the case, the encoding scheme would be weaker and the bias would be detectable by statistical tests. We show empirically in section 5 that publicly available language models are indeed a good fit of the true distribution of natural language.

## A.2 CORRECTNESS

We now assess the adherence of our proposed method to the security model introduced in subsection A.1. Firstly, our method assumes the existence of an underlying symmetric encryption algorithm that is *correct*, *secure*, and whose decryption of a ciphertext $c$ with a key $s$ chosen at random is uniform (DRBG property). We adopt the well-known AES as our cryptographic primitive of choice, but our method can use any symmetric-key algorithm as long as it satisfies the above properties. As for the end-to-end protocol, we can finally observe that:

**Correctness** follows from the correctness of the underlying encryption algorithm, and from the invertible nature of the encoding/decoding scheme (the process is fully deterministic).

**Security** also follows from the security of the underlying encryption algorithm.

**Distributional Deniability** under the assumption that the probability distribution of messages is known explicitly or is represented by a perfect model (as stated in Appendix A.1), this property is satisfied if the distribution of the *encoded plaintext* $e$ corresponding to a random message $m$ is uniform. For a message $m_1$ encoded into $e_1$, this is guaranteed by the ideal model assumption (the model maps the message distribution to the uniform distribution). For a ciphertext decrypted into $e_2$ using a random key $s_2$, the uniformity of $e_2$ is guaranteed by the DRBG property of the underlying encryption algorithm. Finally, if $e_2$ is uniform, decoding $e_2$ into $m_2$ is by definition equivalent to sampling from the probability distribution of $m$ using $e_2$ as a source of entropy. We can thus conclude that the distribution of $m_1$ is indistinguishable from that of $m_2$.

### A.3 PADDING STRATEGIES

**Optimal length** As mentioned, padding is a crucial aspect in ensuring detection is unlikely when a wrong key is used. If a message terminator is expected, it is important to pad the message to a sufficient length such that the terminator always appears in the decrypted message. A simple strategy involves agreeing on a message terminator (e.g. special `<end-of-text>` token, newline, period, comma) – which can be part of the algorithm or even transmitted as part of the ciphertext – and empirically finding the minimum length in which the terminator appears in a given portion (e.g. 99%) of randomly sampled messages, essentially trading off computation/memory for a desired false-positive rate. For reference, we show the empirical distribution for a set of potential terminators in Figure 6 (in the unconditional case). We observe that the `<end-of-text>` token is often not a suitable choice, as it appears sparingly and sometimes exceeds the maximum length supported by the model (at least on GPT2). Other terminators (period, newline) are however more suitable.

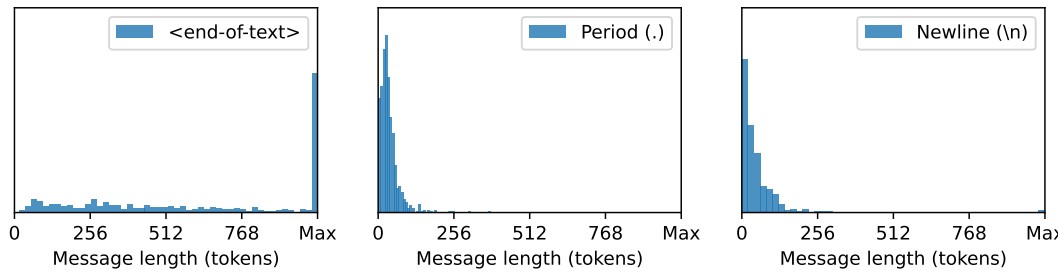

Figure 6: Distribution of message lengths on GPT2 according to different sentence terminators (special end-of-text token, period, and newline), when sampling from the model in an unbiased fashion and without prompt.

**Prompting** Another solution, which can be combined with the aforementioned one, involves prompting. For instance, it is possible to specify a target length as part of the prompt (e.g. a maximum number of characters or words), or even specify the terminator that must be used. Other techniques in a similar spirit, albeit more hardcoded, involve promoting certain tokens during the sampling mechanism to bias the generation process toward shorter sentences, although such a bias may facilitate detection using statistical tests.

### A.4 STATISTICAL TESTS

In this section, we provide more information about our proposed *perplexity test*. For a message consisting of $N$ tokens (where each token is indexed sequentially by a position $i$, and across the vocabulary by an index $j$), we obtain a sequence of discrete probability distributions $\mathbf{p}^{[i]}$ by autoregressively feeding each token through the model. Each token $j$ in a given position $i$ has an associated information content $I^{[i]} = -\log_2 p_j^{[i]}$, which describes the number of bits (or more formally, Shannons) needed for optimally encoding the token given its relative probability. It follows that the total information content of the message can be computed by summing along the length dimension: $I_{\text{total}} = \sum_i^N I^{[i]} = -\sum_i^N \log_2 p_j^{[i]}$. While it is clear that each individual distribution $\mathbf{p}^{[i]}$ is a discrete distribution that depends on the output of the model, for a sequence of $N$ tokens

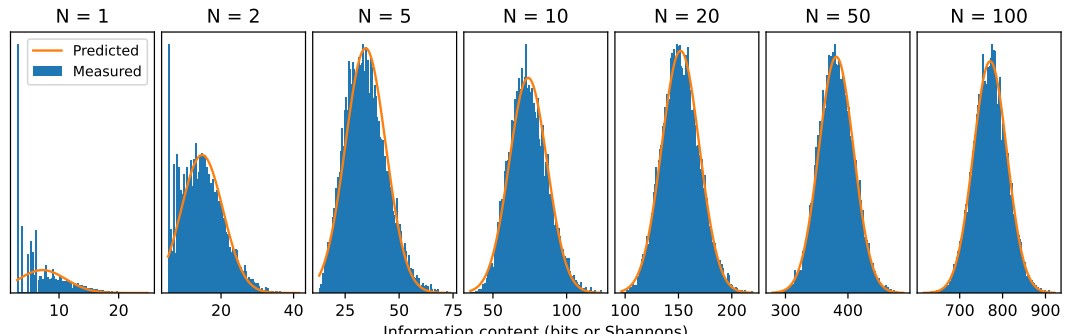

Figure 7: Distribution of the total information content for messages of varying length sampled during a simulation. We show both the empirical distribution (histograms) and our prediction obtained by estimating the parameters of the associated Gaussian distribution.

and a white-box knowledge of each probability distribution predicted by the model, can we learn more about the distribution of $I_{\text{total}}$? As an example, consider Figure 7, where the distribution of the information content as a function of the message length is depicted. In this example, we sample a large number of messages from a sequence of simulated distributions with entropies similar to those encountered in natural language. Although the individual distributions are non-Gaussian, as $N$ increases we observe that the distribution of the sum converges to a Gaussian, which is in line with the central limit theorem. We can analytically estimate its parameters as follows:

$$\mu^{[i]} = -\sum_{h}^{i}\sum_{k}^{M} p_{k}^{[h]} \log_2 p_{k}^{[h]} \qquad \sigma^{2[i]} = \sum_{h}^{i}\sum_{k}^{M} p_{k}^{[h]} \left( -\log_2 p_{k}^{[h]} - \mu^{[h]} \right)^2$$

Afterwards, we can run a two-tailed test to test the null hypothesis that the information content $I_{\text{total}}$ of the sequence under consideration was sampled from $\mathcal{N}(\mu^{[N]}, \sigma^{2[N]})$ with a significance level $p$.

We would also like to highlight that the above formulation assumes that the distribution is stationary across positions $i$, which is not the case for language models. For a better estimate, a possibility is to sample a large number of trajectories and average their statistics, which however makes the test less effective in real-world scenarios. For instance, for a multilingual model, it is expected that different languages (e.g. English and Chinese) have varying perplexities, and averaging them would render the test less specific. Our formulation can be regarded as a contextualized variant of the test where each distribution is conditioned on previous tokens.

## A.5 EXPERIMENTAL SETUP

The datasets used for our experiments (IMDb reviews and Twitter messages), are available at https://huggingface.co/datasets/imdb and https://huggingface.co/datasets/carblacac/twitter-sentiment-analysis. For our analysis, we consider 1000 samples from the training data. We also observe that the Twitter dataset is predominantly characterized by samples with short sequences, leading to a small number of tokens after tokenization, whereas IMDb reviews tend to be longer.

To efficiently compute Huffman codes, we provide a *C++* implementation based on the efficient algorithm proposed in Van Leeuwen (1976). The algorithm's improved efficiency not only expedites the encoding and decoding processes but also contributes to resource optimization.

## A.6 ADDITIONAL QUALITATIVE EXAMPLES

We provide, supplementary to Table 2, more qualitative results when decrypting messages using the wrong key in Table 3.

| Prompt | Correct Message | Decoding 1 | Decoding 2 | Decoding 3 |
|---|---|---|---|---|
| We can meet as always at Amsterdam square. Lets try to make it at: | 11 am. | 2:30pm on the dot. | Sunday, 15.00h | 12:00pm sharp |
| Here is the password for my account: | LeonardiDaVinci | 123456 | 123456789 | 1C-666-573 |
| I have some insider information of which stock to buy: | Apple | AMC. | AAPL and AMD. | SUNE. |
| I finally got a great idea for my research on | how to perform plausible deniability using LLMs. | 18th century French cuisine. | 500px.com. It's a website for photographers. | 3D Printing and decided to start a blog. |
| Today morning at work I | was working my side project I told you about | 1st heard that the volcano in Iceland had erupted. | finished reading a book | managed to figure out how to use my computer |
| The launch of the new IPhone 16 will be made on | Friday the 28th of October. | 8th March 2023. | September 7th. | the same day as the Apple event. |

Table 3: Additional qualitative examples of decoding messages that were decrypted with the wrong keys.

