# OpenReview forum: "Plausibly Deniable Encryption with Large Language Models"
_ICLR.cc/2024/Conference — Submitted to ICLR 2024_

### Official Review · Reviewer_n1KU · 2023-10-31

**Soundness:** 3 good
**Presentation:** 4 excellent
**Contribution:** 4 excellent
**Rating:** 8
**Confidence:** 4

**Summary:**

Plausibly deniable encryption allows to open encrypted messages with multiple keys, providing valid decryptions for each key. This is used as a measure to protect against adversarial parties even when they are able to steal a key, as obtaining a coherent decrypted message does ensure it is the correct one.

Current schemes of plausible deniable encryption are either prohibitively expensive or only allow the decryption of a fixed set of keys. In the latter, the proposed defense is limited as it only increases the space of solutions by a constant factor.

The current work overcomes the limitations of previous approaches by providing a technique that allows to open plausible messages for each possible key of the encryption scheme in an efficient manner. This is achieved by combining encryption, compression techniques used in large language models and Huffman codes. The work evaluates which parameters impact in the probability of detection of correct messages. Under certain conditions, the proposed scheme shows resilience to frequency, correlation and information theoretical tests that attempt to differentiate between true and decoy messages.

**Strengths:**

The paper presents a strong contribution:

- The proposed scheme is original, leveraging the potential of LLMs and information theory in a creative manner. The link between information compression and security has a large potential.

- The authors did a very good job explaining key concepts of their contribution.

- The paper presents an extensive evaluation, providing a clear picture of the situations in which the proposed scheme is applicable and the situations in which is not. Results show that encoded messages are undetected by statistical tests in many realistic scenarios.

- The impact of the contribution is well motivated. The security of encrypted information exchanges benefit from a substantial enhancement if eavesdroppers are not sure that they have decrypted the correct message even having the key.

**Weaknesses:**

A fairly minor weakness that I see is the definition of security of the scheme. The properties outlined in Section 4 to avoid detection seem to make sense, but I think that property (1) could be more precise. For example in regular encryption, ciphertexts must be computationally random, i.e., no polynomial time adversary must be able to distinguish between real noise and the ciphertext. I think such a strong concept of random numbers are not needed for the encoding level of the scheme. However, I get the impression that frequency and correlation tests fit well to evaluate general purpose PRGs, but may fail into assess the lack of a stronger level of randomness.

**Questions:**

As mentioned above, I think the clarity on the detection aspect pointed above is fairly minor but I would be interested to know if the authors have additional insights on how strongly random the encodings should look.

---

> ### Author Response · Authors · 2023-11-19
>
> We thank the reviewer for their careful reading and valuable feedback, to which we respond below.
>
> We acknowledge that the introduction of a precise security model will provide more clarity on the properties of deniability we are interested in. We point the reviewer to our global response and the updated sections A.1 and A.2 in the Appendix for more details.
>
> Regarding the statistical tests, we indeed agree that they cannot fully assess the randomness of the encoded messages. We however decided to include them as a useful tool to verify empirically the assumptions we make regarding the goodness of fit of the model. In the Appendix A.2, we characterize the properties of these encodings more precisely.
>
> > how strongly random the encodings should look
>
> This mainly depends on two factors: (i) whether the model fits well the domain under consideration (or, phrased differently, whether the messages to encode are properly matched to the probability distribution represented by the model), and (ii) in case of a mismatch, what the length of messages is. Longer messages might lead to easier detection if there is a bias in the data or model. Nonetheless, we evaluated this empirically in the paper (Figure 4) and observed no detectable bias even with messages longer than 500 tokens. If there is no mismatch (e.g. in the case of a toy problem, such as sampling from a multinomial distribution that can be quantized exactly), then the encoder will exactly map the sample space of the original distribution to the sample space of the uniform distribution, and the encodings will be as strong as the source of entropy used to sample the initial messages.
>
> We hope that we addressed the reviewer’s concerns, and are more than happy to engage in further discussion.

---

> > ### Comment · Reviewer_n1KU · 2023-11-23
> >
> > Dear authors,
> >
> > Thank you for your response. I appreciate your effort in the addition of the new appendices. My concerns have been properly addressed in your rebuttal and my score remains unchanged.

---

### Official Review · Reviewer_ydp2 · 2023-11-04

**Soundness:** 2 fair
**Presentation:** 2 fair
**Contribution:** 2 fair
**Rating:** 5
**Confidence:** 3

**Summary:**

This paper proposed plausible deniability encryption, which can provide a plausible message instead of actual plaintext when a wrong secret key is used for decryption. In reality, the standard block cipher AES is used as the encryption scheme, and the expected answer is generated using LM with the prompt to guide the desired plausible message.

**Strengths:**

It is interesting that they propose a new cryptographic primitive by mixing the standard encryption method and language models.

**Weaknesses:**

However, by using the encryption algorithm as a simple module, many hints may actually be provided to the attacker at the interface between them.

This assumes that LM also operates like an encryption scheme. In fact, the core to creating deniable encryption lies in LM, so AES operates only as an output encryptor in the proposed scheme. Thus, it appears to rely only on LM when analyzing the security or characteristics of actual deniable encryption. Excluding AES, the evidence for whether the proposed LM is a reliable encryption method appears to be weak other than the assumption that it "looks random."

**Questions:**

Does the proposed method assume a scenario that can be applied in practice? It seems that an attacker would also be able to see that EOS was broken, and thus he/she realizes that the wrong key was used and the output message is not real.
What is the reason for saying encoded plaintext is indistinguishable? Can the encoding be considered an encryption scheme? So why do you need AES?

**Details Of Ethics Concerns:**

I have no concern.

---

> ### Author Response · Authors · 2023-11-19
>
> We thank the reviewer for their feedback, to which we respond below.
>
> Firstly, we point the reviewer to our global response about the definition of deniable encryption that we are adopting and the properties therein. We hope this already makes things more clear. As the reviewer points out, we rely on AES as an underlying primitive to guarantee the security of our algorithm, but we would like to highlight that analyzing the behavior of our method requires a unified view where both the language-model encoder and the underlying encryption algorithm (AES in our case) are considered as a single “block”. In our work, AES is not merely used as an output encryptor, but also as an entropy generator when the message is decrypted using a random key (i.e. a DRBG), which results in the generation of a random plausible message. In this scope, the LM-based encoder/decoder is simply used to map the distribution of real messages to/from the uniform distribution, and not as an encryptor. We have made this more clear in the updated PDF, more specifically in Appendix A.1 and A.2, where we formalize our security scheme explicitly.
>
> > attacker would also be able to see that EOS was broken
>
> This is indeed a very important issue and something we considered by design, and for which we devised a principled padding strategy to avoid precisely this detection mechanism. We discuss this aspect in the first paragraph of Section 3.5 (Implementation details).
>
> > Can the encoding be considered an encryption scheme? So why do you need AES?
>
> As mentioned above, the encoding is not an encryption scheme. Since we assume a white-box model (where the model weights are known to all parties as well as the attacker), the encoding/decoding operations are simply used to deterministically transform one distribution into another (in our case, the message distribution to/from the uniform distribution produced by AES decryption). Furthermore, the encoding itself is not parameterized by a key, so it needs to be paired with a shared-key encryption primitive.
>
> We hope that we addressed the reviewer’s concerns, and are more than happy to engage in further discussion.

---

> > ### Comment · Reviewer_xago · 2023-11-22
> >
> > More than detecting whether EOS is broken, most AES use is authenticated, and decrypting with the wrong key would fail completely. I'm not sure how the scheme proposes to deal with this - the overlayed encryption would have to be CPA secure. Deniability and integrity are very much in tension, in a way the proposal doesn't seem to be able to notice, much less handle. Indeed, in the security model adopted in A1, it seems like a malleated message would decrypt to some plausible-but-incorrect plaintext without this being detectable. Could an adversary exploit this to force a particular message to be received? Maybe not explicitly, but maybe an adversary could try a bunch of perturbations to see which lead to the most "useful" messages. Without a clear, formal security model and proof (which A1/A2 do not provide), it is hard to say whether there are more carefully thought-out attacks along these lines. This is why cryptography demands such formalism.

---

> > > ### Author Response · Authors · 2023-11-22
> > >
> > > We indeed adopt a non-authenticated mode of AES. We also make sure to pad with random bits (so that it is not possible to detect that padding bits are broken when decrypting with the wrong key).
> > >
> > > Nonetheless, we agree with the overarching observation that there might exist attack vectors we did not anticipate (as is the case with many cryptographic primitives that were shown to be vulnerable and were subsequently perfected over time). Our goal with this preliminary study is to show that modern deep learning techniques can benefit applications in cryptography, but there is of course room for improvement as part of future work.

---

### Official Review · Reviewer_xago · 2023-11-12

**Soundness:** 1 poor
**Presentation:** 3 good
**Contribution:** 1 poor
**Rating:** 3
**Confidence:** 4

**Summary:**

This paper proposes the use of LLMs as an encoder/decoder to add a layer of plausible message deniability on top of ordinary encryption. The idea is to make use of the entropy of an incorrect key choice to provide the randomness needed for sampling from the LLM to generate plausible messages without expanding the ciphertext that must be transmitted. It presents experimental results show that incorrect keys do in fact sample to plausible messages.

**Strengths:**

The core idea of this paper is very interesting: LLMs might provide plausible alternative messages given the entropy of an other-than-intended key in a way that reduces the need to send message alternatives or cover traffic or otherwise steganographically encode alternatives in the real message. Presumably, the LLM could be treated as a kind of "shared setup" parameter in the usual cryptographic formalism, and even large values of this kind are regularly considered as part of the cryptography literature. The experimental results showing the positive relationships between sampling strategies for various models are also suggestive that this strategy can be made to work. And some of the ideas about how to build sampling strategies and deal with, e.g., quantizing the distributions that must be sampled from seem useful and helpful.

**Weaknesses:**

I would expect a paper on a cryptography topic - or really any topic in information security - to provide a concrete threat model or security model. In the cryptography literature, these are generally formal and mathematical - indeed, the paper _cites_ the relevant formalism near the end of Section 2. However, this work does not present any such model. Without one, it is unreasonable to make the sort of security claims that the paper makes, as all such claims are relative to _some_ security model. For example, deniability means more than simply extracting a plausible message distinct to the intended message: it should be difficult for an adversary to tell whether the message they received came from the "real" key or an alternative key and whether the message came from the "real" message generation process or a fake one. I didn't see any claims of this sort at all in the latter part of the paper, only claims about the experimental results. Without some kind of theoretical claim on the security of the system ("security" here meaning "according to a goal and set of threats specified by the security model"), it doesn't seem possible to claim that the system provides deniability or improves security (or even doesn't harm it!) in any specific use case.

The rejoinder to this could be that the paper implicitly adopts the formalism of Canetti et al. ,'97. But in fact the approach taken is very different: in this work, the encryption algorithm itself provides the "fake" random choices through the supply of a decoy key. Is this detectable by an adversary? It is not shown here that it is not.

A much more minor point, but an important and closely related one: much is made in the argument against detection of the fact that the encoder's compression capability is good, in the sense that an encoded data stream should be indistinguishable from random in a frequency and correlation sense. In fact, it is a simple enough theorem in cryptography to be assigned as a problem on an undergraduate problem set that there exists a function which passes _any_ battery of tests for randomness and yet does not meet the formal requirements of a pseudorandom generator, in the sense that an adversary can perfectly predict its outputs given enough observations of its behavior. Statistical tests have a one-sided error here, but are being relied on for the wrong "direction" of the security argument (i.e., _failing_ the statistical tests would cause us to _reject_ the security of this scheme, but _passing_ them is not an argument _for_ such security). Indeed, the claim being "tested" here, that the encoded data stream be "indistinguishable from white noise" is a claim that the encoder itself is a semantically secure encryption scheme! (It very clearly is not - knowledge of the model used conveys a lot of information that could be used to distinguish the encodings of different "true" messages).

Lastly and an extremely minor point: Canetti's first name is "Ran", not "Rein" as the reference has it.

**Questions:**

Can the paper provide a clear security model? By this, I don't mean there has to be a mathematical definition of deniability and a proof the scheme meets it (although that would be nice). But it would be good at a minimum to say what an adversary's goals, capabilities, and limitations are in enough detail that claims about the detection of alternative messages can be evaluated for security other than by way of examples.

---

> ### Author Response · Authors · 2023-11-19
>
> We thank the reviewer for the valuable feedback, and we truly believe this has helped us improve our work.
>
> > Security model
>
> Defining a security model (an aspect we initially overlooked in order to present the paper to a broader audience) would indeed be important, therefore we decided to formalize it explicitly. We point the reviewer to the updated paper (Appendix A.1 and A.2) and our global response for more details. In fact, as the reviewer correctly points out, we do adopt the formalism of Canetti et al. '97, and focus on the shared-key deniability scheme (Definition 4). While the definition of correctness and security are mostly unaffected, we adopt a slightly different setting when it comes to the deniability property (see global answer and Appendix A.1). In Appendix A.2, we further explain why our approach adheres to the proposed security model.
>
> > Statistical tests
>
> The reviewer correctly points out that statistical tests are one-sided. We might have miscommunicated the purpose of these tests, which are not meant to show the correctness of the algorithm -- or that the properties of correctness, security, and deniability are satisfied -- but rather to provide a sanity check of the assumptions according to the data/model we use.
>
> > References
>
> We have fixed this.
>
> We hope that we addressed the reviewer’s concerns, and are more than happy to engage in further discussion.

---

### Official Review · Reviewer_jZiK · 2023-11-19

**Soundness:** 3 good
**Presentation:** 3 good
**Contribution:** 2 fair
**Rating:** 5
**Confidence:** 5

**Summary:**

The paper shows how to use (generative) language models to compress and decompress natural sentences, in a way that the encoded (compressed) strings are almost fully using the entropy bound. Hence, a random junk codeword can also be decoded into a random looking sentence.

The idea is to deterministically encode each token $T_i$ with a number $t_i$ in such a way that $t_i$ would allow us to pick $T_i$ again if we had recovered the previous tokens already based on the CDF of the $i$th token's distribution.

The paper then relies on this idea to give an application to "deniable encryption" as follows. We encrypt a message m but first compressing it into a string of numbers $t_1,...$ and then encrypting these strings using standard encryption. Now, if we use a fake key, we end up with a random sequence $s_1,s_2,\dots$ instead, which will lead to another generated (natural looking) sentence.

The paper then discusses some extensions, e.g., to use prompts to contextualize the message. For example if the message is a date, the prompt starts with something that guides the generated message to be a date.

Finally, the paper does some statistical tests to see how indistinguishable are the original texts from the fake decoded variants, and concludes that hey are closely distribution but not fully the same.

**Strengths:**

The strength is to find a nice application to the deniable encryption setting, using the "compressing/decompressing" capability of LLMs.

**Weaknesses:**

The paper's main application is a crypto application. This means the paper needs to be much more formal about its claims, yet the paper does not even have a formal definition of deniable encryption.

In my view, the core contribution of the paper is to derive compression/decompression techniques based on LLMs.
Yet, there seems to be older works on using LLMs for compressing language to its core entropy (eg., this work  from two years ago:
https://www.semanticscholar.org/paper/Lossless-text-compression-using-GPT-2-language-and-Rahman-Hamada/cde63fb5a385fc209107944c6fe19b2d618c407c

**Questions:**

Do you know need any formal properties from the underlying crypto encryption scheme? You say you use AES but I wonder what is needed at abstract level. It seems the scheme itself should have some form of deniability built in so that a random key allows decryption into a legitimate-looking string.

---

> ### Author Response · Authors · 2023-11-19
>
> We thank the reviewer for their valuable feedback.
>
> Since it was also requested by other reviewers, we have already updated the paper with a precise definition of the security scheme in the Appendix A.1. We also refer the reviewer to the global response, where we explain our main changes.
>
> As for compression using language models, indeed there are a plethora of existing works in this area which date back to unigram and n-gram models. After all, language models are explicit probabilistic models whose predictions can be used to guide a “traditional” compressor (LLMs are simply better at this). Therefore, we did not claim the compression idea itself as a contribution, although many of the design choices we made (e.g. working with quantized probability distributions, base encoding scheme) are tailored to our use case. Our contribution lies in the overall protocol as well as its analysis. Nonetheless, we updated the paper to include additional references, including the provided one (but we observe that the latter seems to mostly make use of the tokenizer rather than the model itself).
>
> > Do you need any formal properties from the underlying crypto encryption scheme? You say you use AES but I wonder what is needed at abstract level. It seems the scheme itself should have some form of deniability built in so that a random key allows decryption into a legitimate-looking string.
>
> We have also clarified this in the updated paper (see Appendix A.2). At the abstract level, the underlying encryption algorithm needs to be correct (an encrypted ciphertext should be correctly decrypted into the original message), secure to eavesdropping (the ciphertext must look random), and act as a deterministic random bit generator (DRBG) when the ciphertext is decrypted using a random key. We adopt AES as it is believed to satisfy these properties (in fact, some cryptographic random number generators use AES as a primitive), but in principle, our method can use any primitive that satisfies these properties. The underlying primitive does not require any built-in deniability (but must however not have a built-in integrity check), as this is provided by the model itself and the associated decoding algorithm.
>
> We hope that we addressed the reviewer’s concerns, and are more than happy to engage in further discussion.

---

> > ### Comment · Reviewer_jZiK · 2023-11-22
> > **Re:**
> >
> > Thanks for taking an aim to formalize the notion of deniability that you achieve. As you can see, once one tries to formalize this, there might be surprises (e.g., your notion is weaker than that of Canneti et al., who introduced the notion).
> >
> > I highly suggest that you further compare your notion with theirs and give a new name to your notion, as your notion is *not* the same as the standard one, and this needs to be reflected in what you call your scheme (e.g., distributionally/weakly undeniable?)
> >
> > Also, I would be thankful if you could comment on the paper that I linked above. As I said, your main contribution seems to be using LLMs to compress strings, and this seems to be not new. Please correct me if I am wrong.

---

> > > ### Author Response · Authors · 2023-11-22
> > >
> > > Our definition is indeed different, but we believe it is not necessarily weaker. Plausible deniability has been formulated/implemented in different ways throughout the literature, e.g. steganography (hiding a message inside an image or another medium), schemes where it is clear that a deniable protocol is used but whose content can be denied, and schemes that introduce redundancy. Every protocol comes with its own set of trade-offs, including ours. For example, as we explain in Appendix A.1, we cannot encode an arbitrary decoy message (which would result in a specific key according to Canetti et al.'s formulation), but our method does not introduce redundancy and can be used with any decoy key. The lack of such redundancy means that the receiver cannot be coerced into providing all possible keys (e.g. some deniable encryption protocols assume the existence of a finite number of keys), as any key can be used.
> > >
> > > We agree about using another term for "Deniability" to avoid confusion. We renamed it to "Distributional deniability" in A.1 and A.2. and updated the PDF.
> > >
> > > As for the referenced paper, we believe it is using the BPE tokenizer from GPT-2 to compress the input (which is then fed to a dictionary-based Huffman coder), but not the actual probability distributions predicted by the model (which would make the approach fundamentally different). Nonetheless, we agree that using LLMs for compression has been explored in the past and we do not claim it as a contribution. Our contribution lies in our formulation of the deniable encryption scheme and the corresponding analysis.

---

> > > > ### Comment · Reviewer_jZiK · 2023-11-23
> > > > **Re:**
> > > >
> > > > Thanks for your response and being open to rename your notion. The reason that names do matter in the crypto literature, and if a previous work that is widely referenced uses a name, it is better to use your terminology in a way that does not confuse the reader into thinking that you do achieve the same security notion.
> > > >
> > > > Since you also agree that compression using LLM has been done before, then why is not it that any compression scheme (that encodes natural sentences to close-to their optimal entropy bound) can be used? Just take a string s, encode it into string t, and encrypt t using a one-time pad type encryption (that XORs an inflated key with the message). Then, any perturbed key will decrypt the ciphertext to another string t' that can be decoded to another natural sentence. If you agree that compression is not your novelty, it seems the whole paper's idea is what I wrote in this paragraph, no? Please let me know if I am missing something.
> > > >
> > > > Also, I add that if the main application here is an encryption algorithm, it is much better to be submitted to a cryptography venue to get the proper scrutiny that is highly needed for a new encryption scheme.

---

### Official Review · Reviewer_dPno · 2023-11-22

**Soundness:** 1 poor
**Presentation:** 3 good
**Contribution:** 2 fair
**Rating:** 3
**Confidence:** 4

**Summary:**

This paper investigates the possibility of using a large langage models (LLMs) to provide plausible deniability for conventional encryption schemes in which the objective is to provide a user with the possibility of opening a particular ciphertext under a different key. More precisely, the objective is first to use the LLM to produce a low-level representation of the plaintext before encrypting it using a classical encryption scheme such as AES.

**Strengths:**

-The idea of using a large language model to provide plausible deniability is highly original and provides an interesting example of the combination of recent advances in machine learning with cryptography. More precisely, using the compression capability of LLMs seem interesting approach to be able to achieve plausible deniability.
-The paper is well-written and the authors have done a good job at introducing the necessary background on encoding and decoding.
-Beyond the use of LLMs, the proposed approach is also tested on ImageGPT, which demonstrates the wide applicability of the proposed approach.

**Weaknesses:**

-A detailed characterization on whether the encoding/decoding part can possibly cause a difference on the plaintext in the « normal » situation in which the encryption is performed normally is currently missing from the paper. For instance, what happens if the value of k is less than 32, does it mean that the decoding will result in a different message with a high probability?
-The security analysis should also be more detailed. For instance, it is not clear what are « the optimality assumptions » mentioned in the paper that makes the ciphertext indistinguishable from white noise. Overall, the paper lacks a detailed proof of the security of the proposed scheme. In addition, it also lacks as a review of the main existing families of definitions of the concept of plausible deniability and a detailed discussion on how the proposed notion compares to this.
-The transmission of prompting as as external unencrypted information alongside the ciphertext seems to defeat the purpose of plausible deniability as it will directly indicate to the adversary that there is a tentative to generate a message providing plausible deniability.
-The frequency and correlation tests that are proposed to evaluate the random aspect of the encoded string may not be sufficient to provide a level of security that is required in a cryptographic setting. If possible the authors should clarify whether such test are sufficient to assess the quality and security of cryptographic random number generators.
-It seems that even when the decoding is performed with a different key that a lot of the semantics is preserved (for instance an address is still decoded towards an address). This seems to result in the possibility for an adversary to infer some significant information about the cleartext, which is in contrast to other proposal for deniable encryption. This might be due the information contains in the prompt, which significantly contraints the possible decryption but still this leads to a non-trivial leakage.

**Questions:**

-What does a consensus on a model means in practice (introduction)?
-See also the main issues raised in the weaknesses section.

---

> ### Author Response · Authors · 2023-11-22
>
> We thank the reviewer for their detailed review.
>
> We would like to draw the attention of the reviewer to the newly updated section Appendix A.1, where we characterize our security scheme and our set of assumptions more precisely. Furthermore, in Appendix A.2 we explain why our method adheres to the proposed scheme. We also provide more context in the global answer.
>
> We address individual questions below:
>
> > What does a consensus on a model means in practice (introduction)?
>
> The parties need to agree on the same model (architecture and weights), so that the encoding/decoding process is fully deterministic and correct (the receiver can reliably decrypt the ciphertext into the original message intended by the sender).
>
> > A detailed characterization on whether the encoding/decoding part can possibly cause a difference on the plaintext in the « normal » situation in which the encryption is performed normally is currently missing from the paper. For instance, what happens if the value of k is less than 32, does it mean that the decoding will result in a different message with a high probability?
>
> A key property of our method is correctness (we clarified this in Appendix A.1). If the message is decrypted with the correct key, it can always be decoded exactly (the encoding/decoding process is deterministic and lossless). The value of k describes the quantization granularity of the output probability distribution, and can be considered part of the protocol. As long as the parties agree on the same value of k, the decoding will be correct. The only issue with low values of k is the introduction of quantization errors (which might bias the model and make detection easier), but this is not a problem in the Huffman variant as it does not rely on a choice of k. We also refer the reviewer to the section on implementation details, where we describe engineering considerations to ensure that the model encoding/decoding process is deterministic and thus correct (e.g. quantization).
>
> > The security analysis should also be more detailed. For instance, it is not clear what are « the optimality assumptions » mentioned in the paper that makes the ciphertext indistinguishable from white noise. Overall, the paper lacks a detailed proof of the security of the proposed scheme. In addition, it also lacks as a review of the main existing families of definitions of the concept of plausible deniability and a detailed discussion on how the proposed notion compares to this.
>
> We agree and have formalized our security model and assumptions in detail in the Appendix A.1. In the same section, we also describe our main differences w.r.t. other deniability formulations (notably Canetti et al. ‘97). Finally, we discuss proofs of our security model in the Appendix A.2.
>
> > The transmission of prompting as as external unencrypted information alongside the ciphertext seems to defeat the purpose of plausible deniability as it will directly indicate to the adversary that there is a tentative to generate a message providing plausible deniability.
>
> Indeed, prompting would provide context regarding the topic of the message, but would still provide some level of deniability regarding its content (e.g. if the attacker knows that the receiver is expected to receive an address of a meeting and coerces the receiver into providing a key, the receiver can provide a key that generates a fictitious address). Nonetheless, prompting is an optional feature (the approach can also be used in unconditional mode). We would also like to mention that attempting to generate a message providing plausible deniability (where the attacker has a white-box knowledge that such a protocol is being used) is still a valid scenario, as the content of the original message can be denied. Except for steganography (where the content can be denied entirely), knowledge that a certain protocol is being used is a realistic assumption shared by many other schemes.
>
> > The frequency and correlation tests that are proposed to evaluate the random aspect of the encoded string may not be sufficient to provide a level of security that is required in a cryptographic setting.
>
> We have clarified in Appendix A.2 and Section 4 (Detection) of the revision that these tests are only meant as a sanity check to test whether the assumptions of the model are realized (i.e. whether the model is a good fit for natural language). In A.2, we have added more formal proofs of correctness that do not depend on such empirical tests.
>
> > It seems that even when the decoding is performed with a different key that a lot of the semantics is preserved (for instance an address is still decoded towards an address)
>
> This is because the text demos in the paper were generated with prompting (the prompt is also shown), but as explained earlier it is also possible to use the model without prompt, which will generate a completely random/unrelated message. By contrast, the demos on ImageGPT were generated unconditionally.

---

### Author Response · Authors · 2023-11-19

We thank the reviewers for the careful reading and the points raised, and are glad to hear that they find the fundamental idea very interesting and original. We respond to reviewers separately to address all of their concerns. Furthermore, we have updated the text based on the suggestions made (changes are highlighted in red in the PDF). In particular, relevant changes can be found in Appendix A.1 (security model), Appendix A.2 (adherence to security model), as well as in Section 3 (method) and Section 4 (detection). We are also open to moving the changes relative to the security model in the main text if the reviewers think it is more appropriate.

The paper is addressed to a non-cryptographic audience, so we did not initially embark on a rigorous formalism of the security model used. As raised by reviewers (in particular xago), we do realize though, that such a formalism will greatly benefit the scientific impact of the paper as a bridge between the cryptographic and machine learning community. We thus specifically outline the security model assumed in our paper in Appendix A.1, and assess its validity in Appendix A.2. As suggested by reviewer xago, we are indeed inspired by the scheme proposed in Canetti et al. ‘97, in particular Definition 4 (Deniable encryption in the shared-key scenario). In the revision, we however highlight that our model of deniability is slightly different from that of Canetti et al. ‘97. In the latter, decoy messages can be arbitrarily chosen, whereas decoy keys are derived through a “faking” function. In our work instead, decoy keys can be arbitrarily chosen, and decoy messages are randomly drawn from the distribution of messages. We explain this difference (and the resulting implications) in Appendix A.1.

---

### Meta-Review · Area_Chair_u8A2 · 2023-12-10

**Metareview:**

The reviewers were split about this paper and did not come to a consensus: on one hand they appreciated the originality of the idea and the extensive evaluation, on the other they had issues with (a), the light assessment of the randomness at encoding level by the use of statistical tests, (b) the possibility that the EOS tokens could break the encoding distribution, (c) some vectors of attack such as exploiting malleability of the encryption, (d) the definition of security, and (e) proof of security. After going through the paper and the discussion I have decided to vote to reject based on the above issues. The authors added more details on the above issues during the rebuttal phase, Appendices A.1 and A.2 are appreciated. Ultimately the final decision hinged on whether a formal definition of security is necessary, particularly given that this is an ML venue. The reviewers disagreed on this point, however, given that the paper is pitched in the abstract as “a novel approach for achieving plausible deniability in cryptography” as well as throughout the paper, a formal proof of security is necessary to make these claims. For this reason I vote to reject. If the authors can add a formal proof of security, and address the other detailed points made by the reviewers, the paper will be an excellent contribution to a top-tier ML conference such as ICLR.

**Justification For Why Not Higher Score:**

The paper makes a security claim without a security proof.

**Justification For Why Not Lower Score:**

N/A

---

### Decision · Program_Chairs · 2024-01-16

Reject